# Antimicrobial Resistance and Virulence Factors Assessment in *Escherichia coli* Isolated from Swine in Italy from 2017 to 2021

**DOI:** 10.3390/pathogens12010112

**Published:** 2023-01-09

**Authors:** Patrizia Bassi, Claudia Bosco, Paolo Bonilauri, Andrea Luppi, Maria Cristina Fontana, Laura Fiorentini, Gianluca Rugna

**Affiliations:** Istituto Zooprofilattico Sperimentale della Lombardia e dell’Emilia Romagna, Via A. Bianchi 9, 25124 Brescia, Italy

**Keywords:** antimicrobial resistance, *Escherichia coli*, ETEC, swine

## Abstract

Prudent antibiotic use in pigs is critical to ensuring animal health and preventing the development of critical resistance. We evaluated the antimicrobial resistance (AMR) pattern in commensal and enterotoxigenic *Escherichia coli* (ETEC) isolates obtained in 2017–2021 from pigs suffering from enteric disorders. Overall, the selected 826 *E. coli* isolates showed the highest level of resistance to ampicillin (95.9%), tetracycline (89.7%), cefazolin (79.3%), and trimethoprim/sulfamethoxazole (74.8%). The resistance rates of the isolates to ampicillin increased (*p* < 0.05), reaching 99.2% of resistant strains in 2021. Regarding isolates harboring virulence genes, ETEC F18+ were significantly more resistant to florfenicol, gentamicin, kanamycin, and trimethoprim/sulfamethoxazole than ETEC F4+ strains. *E. coli* lacking virulence factor genes were more resistant to amoxicillin with clavulanic acid and cefazolin, but less resistant to gentamicin (*p* < 0.01) than isolates harboring virulence factors. Throughout the study period, a significant number of ETEC F18+ isolates developed resistance to florfenicol, gentamicin, and kanamycin. Finally, ETEC 18+ significantly (*p* < 0.05) increased resistance to all the tested antibiotics. In conclusion, AMR varied for *E. coli* over time and showed high levels for molecules widely administered in the swine industry, emphasizing the need for continuous surveillance. The observed differences in AMR between commensal and ETEC isolates may lead to the hypothesis that plasmids carrying virulence genes are also responsible for AMR in *E. coli*, suggesting more research on genetic variation between pathogenic and nonpathogenic *E. coli*.

## 1. Introduction

Antimicrobial Resistance (AMR) is the inability or reduced activity of an antimicrobial agent to inhibit the growth of a microorganism, and for bacteria is referred to as antibiotics. Antibiotics used in human and veterinary medicine belong frequently to the same classes and the use in both, humans and animals, can improve AMR by the selection of resistant clones whether these are pathogenic or commensal [1]. In order to reduce antimicrobial resistance and antimicrobial usage (AMU) in Italy, a national plan to contrast antimicrobial resistance (Piano Nazionale di Contrasto dell’Antimicrobicoresistenza—PNCAR) was implemented in 2017. As reported in the European report on antimicrobial sales in Europe in livestock (2021) [2], a significant reduction (−33.6%) was recorded from 2017 to 2020 in Italy, even if this country still remains one of the main antimicrobial consumers in Europe. AMR in swine production represents an issue mainly for enteric disorders (colibacillosis, salmonellosis, and *Brachyspira* spp. infection). *Escherichia coli* is an important infectious agent responsible for a wide range of diseases in pigs, including neonatal diarrhea (ND), postweaning diarrhea (PWD), edema disease (ED), septicemia, polyserositis, coliform mastitis (CM), and urinary tract infection (UTI). The different pathological manifestations are due to several virulence factors (adhesins and toxins) that can be combined into different “virotypes” [3]. The main adhesins involved in swine enteric pathology are F4 and F18, which, combined with different toxins (STa, STb, LT, and others), can be responsible for ND, PWD, and ED [3]. Particularly, *E. coli* responsible for ND and PWD are classified as enterotoxigenic *E. coli* (ETEC) and can produce a combination of adhesins and toxins as reported in Table 1 [3]. Moreover, *E. coli* are widely diffused in the swine gut and can be part of the normal resident flora. Antibiotic resistant intestine commensal and pathogenic *E. coli* may be selected as a result of antimicrobial oral and injective delivery [3]. The study of phenotypic AMR of commensal ‘indicator’ *E. coli* from the intestinal flora provides information on the reservoirs of resistant bacteria that could potentially be transferred between animal populations and between animals and humans, as well as indirect information on the reservoirs of food resistance genes in animals and food that could be transferred to bacteria that are pathogenic for humans and/or animals [1].

The aim of this observational, retrospective analysis was to investigate (*i*) the AMR rates and temporal pattern in *E. coli* isolates from pigs suffering from enteric disorders in the Emilia Romagna region, Italy, from 2017 to 2021; (*ii*) the differences in AMR in *E. coli* with and without virulence factor genes; (*iii*) the rate and temporal trend of multi drug resistance (MDR).

## 2. Results

A total of 826 *Escherichia coli* isolated from pathological swine samples were assessed contemporaneously for the presence of virulence factor genes and for antimicrobial susceptibility. *E. coli* isolates were collected over five years (2017–2021) by the diagnostic laboratories of the Istituto Zooprofilattico Sperimentale della Lombardia e dell’Emilia Romagna (IZSLER) in the territories of Reggio Emilia (n. 417/826, 50.5%), Parma (n. 329/826, 39.8%), Ravenna (n. 29/826, 3.5%), Modena (n. 27/826, 3.3%), Forlì (n. 19/826, 2.3%), and Bologna (n. 5/826, 0.6%). Antimicrobial susceptibility was investigated for the overall isolates (n. 826) and compared between isolates lacking virulence factors (n. 155/826, 18.8%) and ETEC F18+ (n. 230/826, 27.8%) or ETEC F4+ (n. 144/826, 17.4%) isolates. Additionally, throughout the research period from 2017 to 2021, yearly assessment of the evolution of antimicrobial susceptibility and the spread of multi-resistant isolates were carried out.

### 2.1. Detection of Virulence Factors Genes

Virulence genes were identified in 671/826 (81.2%) isolates. Adhesins encoding genes were mainly F18+ (272/671, 40.5%) and F4+ (170/671, 25.3%), followed by F41+ (43/671, 6.4%), F5+ (30/671, 4.5%), F4+ F18+ (24/671, 3.6%), F4+ F41+ (14/671, 2.1%), F6+ (9/671, 1.3%), F18+ F41+ (1/671, 0.1%), F4+ F6+ (1/671, 0.1%), F4+ F41+ (1/671, 0.1%), and F5+ F6+ (1/671, 0.1%). Out of 671 isolates, 22 (3.3%) were Shiga toxin-producing *Escherichia coli* (STEC) (virotype: F18 Stx2e), 87 (13%) were *E. coli* with one or more adhesins-encoding genes but without toxins-encoding genes, and 105 (15.6%) were isolates with toxins-encoding genes despite the absence of adhesins.

Isolates expressing simultaneously one or more adhesins and toxins encoding genes were classified as ETEC (457/671, 68.1%). The most common ETEC virotypes were F18, STa, STb (136/457, 29.8%); F4, STa, STb (40/457, 8.8%); F4, STa, STb, LT (35/457, 7.7%); F4, STb, LT (32/457, 7%); and F18, STa, STb, LT (30/457, 6.6%) (Appendix A).

The statistical analyses were carried out considering the two most representative groups of ETEC isolates: ETEC F18+ (230/457, 50.3%) and ETEC F4+ (144/457, 31.5%) regardless of the type of enterotoxins produced.

### 2.2. Antimicrobial Susceptibility

#### 2.2.1. AMR Results and Comparison between Virulence-Positive and Negative Isolates

The number of tested strains and percentage of resistance for all the tested antimicrobials are reported in Table 2.

The main resistances were recorded for ampicillin (791/826, 95.9%), followed by tetracycline (740/826, 89.7%), cefazolin (654/826, 79.3%), and trimethoprim + sulfamethoxazole (617/826, 74.8%) (Table 2). For each of the ten antimicrobials taken separately, resistance was observed for over 50% of the tested isolates. Kanamycin and enrofloxacin both displayed the minor resistance (489/826, 57.1% and 471/826, 59.3% resistant isolates, respectively) (Table 2).

Regarding *E. coli* lacking virulence factor genes, resistance to ampicillin was recorded in 153/155 (98.7%) of the isolates (Table 2; Appendix A). Moreover, in comparison with isolates with positive virulence genes, negative *E. coli* showed statistically significant (*p* < 0.01) higher resistances to amoxicillin and clavulanic acid (119/155, 76.8% vs. 416/671, 62%), and cefazolin (136/155, 87.7% vs. 519/671, 77.3%), whereas gentamicin resistance was significantly (*p* < 0.01) lower (76/155, 49% vs. 431/671, 64.2%) (Table 2; Appendix A).

Concerning ETEC, F18+ isolates were significantly (*p* < 0.01) more resistant to trimethoprim + sulfamethoxazole (195/230, 84.8% vs. 88/144, 61.1%), florfenicol (187/230, 81.3% vs. 89/144, 61.8%), gentamicin (184/230, 80% vs. 82/144, 56.9%), and kanamycin (156/230, 67.8% vs. 78/144, 54.2%) than F4+ (Table 2; Appendix A).

#### 2.2.2. Trend in AMR from 2017 to 2021

The percentage of resistant strains to every tested antimicrobial listed from 2017 to 2021 is reported in Appendix A.

From 2017 to 2021, a statistically significant (*p* < 0.05) increase in ampicillin-resistant *E. coli* (93.5–99.2%) was identified (Figure 1, Appendix A). During the study period, a statistically significant (*p* < 0.05) number of ETEC F18+ isolates increased resistance to florfenicol (70–95.8%), gentamicin (70–91.7%), and kanamycin (50–91.7%) (Figure 1, Appendix A); no variations were observed for F4+ isolates (Appendix A).

#### 2.2.3. Multi-Resistant Isolates

Regarding multi-resistance, 18.3% of the isolates showed concurrent resistance to all 10 antimicrobials tested; moreover, 76.5% of the isolates were simultaneously resistant to six or more antimicrobials (multi-resistant isolates) (Table 3).

Only 4/826 (0.5%) of the tested *E. coli* were fully susceptible (Table 3) and none (0/144) of the tested F4+ *E. coli* were fully susceptible (Appendix A).

Even though the percentage of isolates resistant to six or more antibiotics did not change significantly from 2017 to 2021, the trend in multi-resistance of the 826 strains led to a statistically significant (*p* < 0.05) decrease in isolates resistant to only one antibiotic (Table 3). Additionally, for F18+ *E. coli* strains, there was a statistically significant upward trend in current resistance to 10 molecules (Table 3).

For F4+ E. *coli* isolates, no statistically significant changes were observed (Appendix A).

## 3. Discussion

This study is the result of a 5-year retrospective analysis of *E. coli* isolates originating from swine pathological samples in the Emilia Romagna region, Italy. Since most ETEC gene clusters for fimbriae are found on plasmids that also carry enterotoxins and antibiotic resistance genes [4], it is reasonable to assume that *E. coli* harboring various virulence factors will differ in antimicrobial susceptibility.

Globally, a review on AMU in swine concluded that the most commonly used molecules are penicillins and tetracyclines [5]. According to the last report on sales of veterinary antimicrobials in Europe [2], from 2010 to 2016, the highest selling antimicrobial classes were penicillins, tetracyclines, sulfonamides, and polymyxins. During 2016, a governing ban on the production of colistin-associated oral medicaments resulted in a decrease in polymyxins’ utilization in livestock, then sales of macrolides and lincosamides increased. In 2020, the three highest-selling antimicrobial classes were penicillins, tetracyclines, and sulfonamides, accounting for 33.6%, 26.9%, and 14.7%, respectively, of total sales. The overall isolates’ high resistance to ampicillin, tetracycline, and trimethoprim + sulfamethoxazole reported in this study is consistent with the selling EMA report; however, resistance to colistin was not assessed. Other European, Asian, and Americans countries reported similar results on *E. coli* isolates between 2005 and 2018, with higher percentages of resistance to ampicillin [6,7,8,9], tetracycline [6,8,9,10], and trimethoprim associated or not with sulfonamides [8,9,10]. Finally, also in the European report on AMR in zoonotic and indicator bacteria published by the European Food and Safety Authority (EFSA) in 2021 [1], resistance to ampicillin, sulfamethoxazole, trimethoprim, and tetracycline were the most common resistance traits observed in indicator *E. coli* collected at slaughterhouses from broilers, fattening turkeys, fattening pigs, and calves in 2018–2019.

Interestingly, in the present study, resistance to amoxicillin with clavulanic acid and cefazolin was significantly higher for *E. coli* strains lacking virulence factors. On the contrary, the isolates lacking virulence factors showed significantly lower resistance to gentamycin. Additionally, F18+ isolates were more resistant to gentamicin and kanamycin than F4+ isolates.

Numerous genes in *E. coli* encode for resistance to β-lactams; some of them confer resistance only for narrow-spectrum β- lactamase, which can inactivate penicillin and aminopenicillins [11]. Recently, genes that confer resistance to extended spectrum β-lactamase (ESBL) emerged in *E. coli* from humans and animals; additionally, genes coding for carbapenemases have been detected [11]. The dissemination of ESBL among *E. coli* from animals is mainly driven by horizontal gene transfer since the majority of ESBL genes are plasmid-located [11]. Some plasmids that carry ESBL genes seem to be more successful than others; moreover, some plasmids harbor additional resistance genes besides the ESBL genes, which may enable coselection and persistence of ESBL-carrying plasmids, even in the absence of β-lactams selective pressure [11]. Third and fourth generation cephalosporins were not tested in our study, and information about the AMR carrying genes was unavailable. However, the differences in AMR observed between *E. coli* harboring or not virulence factors may lead to the hypothesis that different plasmids are responsible for AMR in *E. coli* depending on the presence of adhesins and toxin-genes, even though more research on genetic variation between pathogenic and nonpathogenic *E. coli* is needed.

Gentamicin is an aminoglycoside used to treat neonatal colibacillosis in piglets from day 1 to day 3 of age [12]. Most clinically important resistance to aminoglycosides is caused by plasmid-mediated enzymes that modify the molecules so they become unable to reach the target site [11]. Possibly, the administration of gentamicin in ND or PWD due to ETEC isolates may have contributed to maintaining a higher level of resistance in isolates harboring virulence genes. In 2013, an Italian study on *E. coli* F4+ isolates showed an increasing trend, even if not statistically significant, of in vitro resistance to all aminoglycosides tested, particularly to apramycin and gentamicin [13]. In our investigation, ETEC F18+ isolates exhibited higher resistance rates to gentamicin and kanamycin than F4+ isolates. However, increased resistance to florfenicol and trimethoprim + sulfamethoxazole was also observed in F18+ isolates, indicating a multi-resistance pattern. Once again, additional research on the genetics of resistance in other ETEC isolates may be helpful to better understand these results. Indeed, it was reported that *E. coli* from pigs may be an important reservoir for the transfer of gentamicin resistance genes or bacteria to humans [14], and the spread of gentamicin resistance in humans is of great concern considering the importance of this antibiotic in human medicine [15].

Concerning the trend of resistance to each tested antibiotic, statistically significant results were identified only for ampicillin, which showed an increased trend from 93.5% of resistant strains in 2017 to 99.2% in 2021. Ampicillin and tetracycline trends are reported in the European report on AMR in zoonotic and indicator bacteria from humans, animals, and food, because those antibiotics are considered the most commonly used in food-producing animals in Europe, and a variation in this trend is thought to primarily reflect changes in the AMU [1]. According to this report, the trend in resistance to ampicillin among *E. coli* indicators isolated from fattening pigs has decreased in five countries (Cyprus, Germany, Netherlands, Portugal, and Switzerland), while it has increased in nine countries (Austria, Belgium, Denmark, France, Ireland, Poland, Romania, Slovakia, and Spain). Even though a 17% drop in AMU was observed in Italy from 2010 to 2020 [2], trends in the resistance of indicator *E. coli* isolated from fattening pigs showed a statistically significant rise in ciprofloxacin resistance, an increase in resistance for ampicillin, and a slight decline in tetracycline resistance, even though not significant in our country [1]. Differences in the target population of sampling may be the cause of discrepancies between our results and the reported data for ampicillin. *E. coli* isolated from pathological samples in this study typically came from piglets or weaners affected by ND and PWD or ED, as opposed to fattening pigs, and in Italy, ampicillin is administered mostly in piglets and weaners [16].

According to the last report on antimicrobial consumption and resistance in Europe, regarding aminopenicillins, a statistically positive association between consumption and resistance in *E. coli* isolated from poultry and pigs was found [17]. Moreover, for *E. coli*, resistance in bacteria isolated from humans was significantly related to resistance in bacteria from food producing animals, even if this correlation can be due to the similar levels of consumption that a country can display in both sectors (humans and animals), so a country with high aminopenicillin consumption in one sector would tend to have a high consumption of aminopenicillins in the other sector [17]. Since ESBL-encoding genes also confer resistance to aminopenicillins, the use of this molecule may select for bacteria harboring these genes; moreover, these genes are usually located in mobile elements that also harbor genes encoding for resistance to other antimicrobials [17]. Therefore, the high levels of aminopenicillin resistance that resulted from this study and the statistically significant increasing trend in resistance for this antimicrobial class are a matter of concern not only for the failure of antimicrobial therapy in swine enteric diseases due to *E. coli* infections but also for the co-selection for *E. coli* producing ESBL that can spread in the swine industry. Moreover, a spread of ESBL-encoding genes via mobile genetic elements can reach zoonotic microorganisms such as *Salmonella* spp. and *E. coli*, which can reach humans through the food chain, even if robust quantitative data are still lacking to conclude that there is real enrichment of the human microbiota with ESBL producing *E. coli* isolates via the food chain [18].

This study does not support the rise in ciprofloxacin resistance that the EFSA reported for Italy from 2009 to 2019 [1]. Instead, it found a slight reduction in enrofloxacin resistance among isolates from 2017 to 2021, which was not statistically significant. Differently, an Italian study on F4+ *E. coli* isolated from pig samples from 2002 to 2011 showed a statistically significant increasing trend of resistance over the whole period to different fluoroquinolones (enrofloxacin, marbofloxacin, danofloxacin) and quinolones (flumequine) [13]. The lower level of resistance to enrofloxacin, a fluoroquinolone considered HPCIA (Highest Priority Critically Important Antimicrobials for Human Medicine), compared to other molecules reported in this study is positive data, even though nalidix acid (quinolone) still has a high resistance value recorded. Various mutations may be responsible for the different levels of resistance determined for quinolones and fluoroquinolones. Resistance of *E. coli* to quinolones and fluoroquinolones is usually due to mutations in the drug targets, primarily the DNA gyrase (*gyrA* and *gyrB*). Single mutations in *gyrA* may confer resistance to quinolones, but further mutations in *gyrA* or in a second enzyme named Topoisomerase IV are required to confer resistance to fluoroquinolones [11]. Additionally, other mechanisms, also plasmid-mediated, may play a role in quinolones and fluoroquinolones resistance, such as reduced permeability, protection of the target structures, or upregulation of efflux pumps [11]. Fortunately, these resistance determinants do not confer a high level of resistance to this antimicrobial class, but rather a lesser susceptibility. However, they might contribute to the selection of isolates with higher levels of resistance through other chromosomally encoded mechanisms. A plasmid-mediated quinolones resistance gene (PMQR), named *oqxAB,* was identified in unrelated *E. coli* isolated from food producing animals and located on different plasmids. Interestingly, PMQR *oqxAB* confers resistance not only to quinolones but also to other drugs such as trimethoprim and chloramphenicol [11].

According to EFSA [1], resistance in food-producing animals can be addressed by considering multidrug resistance (MDR) as well as the proportion of fully susceptible indicator *E. coli* isolates. The latest indicator should reflect the degree of AMU, because only *E. coli* that is rarely, if ever, exposed to antimicrobials will be fully susceptible [19]. In this study, only 0.5% of the tested *E. coli* were fully susceptible, although an EFSA report on AMR in zoonotic and indicator bacteria stated that 12.9% of indicator *E. coli* isolated from fattening pigs were fully susceptible to antibiotics in 2019 [1]. In a study conducted in Denmark about cross- and co-resistance in 765 porcine *E. coli* isolates between 2009 and 2013, 45% of the tested isolates were fully susceptible to all antimicrobials tested [20]. This result was similar to that already described by EFSA in 2109, where 42.1% of indicator *E. coli* isolated from Danish pigs were fully susceptible to antibiotics [1]. In our study, only four (0.5%) *E. coli* isolates were completely susceptible to all ten antimicrobials tested. This difference between results obtained in Denmark and Italy would be related to a different AMU. In the annual sales of veterinary antimicrobial agents for food-producing animals by country from 2010 to 2020, 37.2 mg/PCU of antibiotics were sold in Denmark in 2020, compared to 181.8 mg/PCU of antibiotics sold in Italy in the same year. Even in previous years, the difference in terms of veterinary antimicrobial sales between the two countries was always significant [2]. The main multi-resistant profile in Denmark was resistance to ampicillin (AMP), streptomycin (STR), sulfonamides (SUL), tetracycline (TET), and trimethoprim (TRI) (6% n = 48/765) [20]. Overall, 32% (n = 245) of the tested isolates were multi-resistant using the EFSA definition of resistance to more than five antimicrobial classes. In the present study, 76.5% of the isolates were simultaneously resistant to six or more antibiotics. Between 2017 and 2021, ETEC F18+ resistant to all 10 antibiotics tested showed a statistically significant increasing trend. In the same years, no statistically significant increasing trend was shown by multi-resistant ETEC F4+, contrary to the result obtained in *E. coli* isolated in Italy between 2002 and 2011 [13]. This result may be related to differences in multi-resistant mobile genes associated with virulence factors and suggests promoting molecular research on the role of virulence factors in AMR in *E. coli*. With a genomic analysis of ETEC F4+ and F18+ isolates from post-weaning pigs, researchers in Denmark investigated the possible plasmid location of ETEC fimbriae and toxins producing genes, as well as their potential cooccurrence with AMR determinants. F4 and F18 fimbriae encoding genes were plasmid located in all strains, and in more than 70% of the isolates, toxins were plasmid located too. The majority of the AMR genes were predicted to be plasmid-located, and AMR and virulence genes were often predicted to be part of the same plasmid component [21]. However, further detailed studies are needed to corroborate this, since the program used for predictions (plasmidSPAdes) does not separate plasmids of the same type present in a single strain.

## 4. Materials and Methods

### 4.1. Study Design

Over a five-year period (2017–2021), the presence of *E. coli* strains was investigated in biological materials (feces, fecal swabs, lymph nodes, and intestines) from diarrheic pigs as part of the routine activity of the diagnostic laboratories of IZSLER, Emilia-Romagna, Italy. Inclusion criteria for the retrospective analysis were: (*i*) *E. coli* isolates tested for virulence factor genes; (*ii*) sharing the same panel of antibiotics in the antimicrobial susceptibility test; (*iii*) including prototype molecules for the most classes of antibiotics used in swine health practice.

### 4.2. Isolation of Escherichia Coli

The isolation procedure was consistent during the whole study period. Within 24 h after collection, the samples were processed, plated onto blood agar and McConckey agar, and incubated aerobically at 37 ± 2 °C for 24 h. Presumptive identification of *E. coli* was based on the growth and morphological characteristics. Single colonies were further characterized by standard biochemical tests using Microgen GNA ID system (Microgen Bioproducts, Ltd., Camberley Surrey, UK).

### 4.3. Virulence Genes Characterization

*E. coli* isolates were screened by multiplex PCR for the presence of the major virulence genes of pathogenic *E. coli*, including genes for five different adhesins (F4, F5, F6, F18, and F41) and four different toxins (LT, STaP, STb, and Stx2e), according to the method of Casey and Bosworth [22]. Briefly, DNA was obtained from each *E. coli* isolate using a hot lysis procedure in which the sample was harvested by centrifugation (12,000× *g* for 5 min), washed three times in distilled water, boiled at 97.5 ± 2.5 °C for 10 min, and immediately cooled on ice for 2 min [22]. After centrifugation, the extracted DNA was subjected to multiplex PCR to screen virulence factors (VFs) using specific primers (Appendix A) [22]. The PCR reaction mixtures contained 18 primers at a concentration of 0.5 µmol each, with 0.2 mmol deoxyribonucleotide triphosphate mix, 1× reaction buffer, 5 mmol MgCl_2_, and 2.5 units of Taq polymerase in a final volume of 20 µL [22]. The amplification conditions were as follows: initial denaturation at 94 °C for 10 min, followed by 30 cycles of denaturation for 30 s at 94 °C, annealing at 55 °C for 45 s, and extension for 1.5 min at 72 °C [22]. The extension time was increased by 3 s each cycle, and the final extension was 10 min at 72 °C [22]. The amplification products were then separated and detected by electrophoresis using 4% agarose gels at 80 V for 1 h. [22].

### 4.4. Antimicrobial Susceptibility Test

The antimicrobial susceptibility of all the isolates was assessed on Mueller–Hinton agar using the disc diffusion method for the following ten antimicrobials: nalidixic acid (30 µg), amoxicillin and clavulanic acid (20–10 µg), ampicillin (10 µg), cefazolin (30 µg), enrofloxacin (5 µg), florfenicol (30 µg), gentamicin (10 µg), kanamycin (30 µg), tetracycline (30 µg), and trimethoprim + sulfamethoxazole (1.25–23.75 µg). The *E. coli* were classified as sensitive, intermediate, and resistant, following CLSI interpretative criteria [23]. For the statistical analysis, intermediate *E. coli* isolates were grouped with the resistant isolates. The strains were defined as MDR when they presented resistance to at least six antibiotics.

### 4.5. Statistical Analysis

The antimicrobial resistance rate was calculated for each antibiotic considered in the whole sample (n. 826) and in groups with various fimbrial genes. The comparison between groups was performed with the chi-square test and Fisher’s exact test, and the differences were considered statistically significant at *p* < 0.01. Pearson’s correlation test (Pearson’s r) was used to assess the presence of a trend among years of isolation (weighted for the different number of isolates during the years). Additionally, the multi-resistance for each year has been evaluated. An increasing or decreasing trend of antimicrobial resistance was considered statistically significant at *p* < 0.05. All analyses were performed using Intercooled STATA version 7.0 (StataCorp).

## 5. Conclusions

Even if the role of animals in the transmission of AMR to humans is still debated, the fight against those resistant bacteria needs a One-Health approach. The increasing trend in resistance to certain antimicrobials such as ampicillin which resulted from this study, reaching 99.2% of the isolates, should be taken into consideration as evidence of the wide consumption of this antimicrobial class in these animals. Additionally, the observed trend in multi-resistance among the 826 strains led to a significant (*p* < 0.05) decrease in isolates resistant to only one antibiotic from 2017 to 2021.

Differences in AMR related to the presence of virulence genes were recorded for amoxicillin with clavulanic acid, cefazolin, gentamicin, and florfenicol; moreover, an increasing trend in resistance to aminoglycosides from 2017 to 2021 and an increasing trend in contemporary resistance to more than 10 antimicrobials were assessed for F18+ *E. coli*. Further analysis focused on the role of this and other virulence factors in AMR in pigs is needed.

## Figures and Tables

**Figure 1 pathogens-12-00112-f001:**
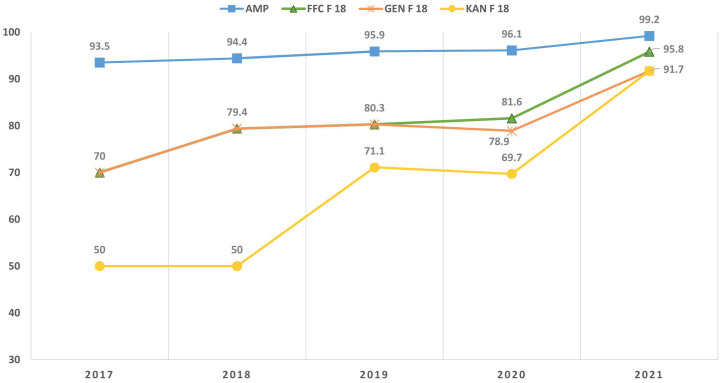
Trend of resistance, from 2017 to 2021, to ampicillin (AMP) of the overall *Escherichia coli* strains (*p* < 0.05) and resistance to florflenicol (FFC), gentamicin (GEN), and kanamycin (KAN) of the enterotoxigenic *Escherichia coli* (ETEC) F 18+ isolates (*p* < 0.05).

**Table 1 pathogens-12-00112-t001:** Different combination of adhesins and toxins of enterotoxigenic *Escherichia coli* (ETEC) responsible for intestinal pathology in pigs [3]. Fimbriae (F), heat-stable toxin a (STa); heat-stable toxin b (STb); heat-labile toxin (LT); Shiga toxin (Stx), enteroaggregative heat-stable toxin (EAST-1), neonatal diarrhea (ND), postweaning diarrhea (PWD).

Virotype		Disease
Adhesins	Toxins	
F5, F6, F41	STa	ND
F4	STa, STb, LT, EAST-1, α-hemolysin	ND
F4, AIDA	STa, STb, LT, EAST-1, α-hemolysin	PWD
F18, AIDA	STa, STb, LT, Stx (VT) EAST-1, α-hemolysin	PWD

**Table 2 pathogens-12-00112-t002:** Number and percentage of *E. coli* resistant to the tested antimicrobials. *E. coli* isolates tested in total (n. 826), virulence genes positive and negative isolates (n. 671 n. 155) and enterotoxigenic *Escherichia coli* (ETEC) F4+ (n. 144) and ETEC F18+ (n. 230). Nalidixic acid (NA), amoxicillin and clavulanic acid (AMC), ampicillin (AMP), cefazolin (CZ), enrofloxacin (ENR), florfenicol (FFC), gentamicin (GEN), kanamycin (KAN), tetracycline (TET) trimethoprim + sulfamethoxazole (SXT).

Antibiotics	Overall Isolates	Isolates Positive for Virulence Genes	Isolates Negative for Virulence Genes	ETEC F4+	ETEC F18+
	N° (%)	N° (%)	N° (%)	N° (%)	N° (%)
NA	598 (72.4)	497 (74.1)	101 (65.2)	113 (78.5)	178 (77.4)
AMC	534 (64.8)	416 (62) *	119 (76.8) *	87 (60.4)	137 (59.6)
AMP	791 (95.9)	639 (95.2)	153 (98.7)	137 (95.1)	219 (95.2)
CZ	654 (79.3)	519 (77.3) *	136 (87.7) *	116 (80.6)	177 (77)
ENR	471 (57.1)	386 (57.5)	86 (55.5)	92 (63.9)	120 (52.2)
FFC	527 (63.9)	443 (66)	85 (54.8)	89 (61.8) *	187 (81.3) *
GEN	506 (61.4)	431 (64.2) *	76 (49) *	82 (56.9) *	184 (80) *
KAN	489 (59.3)	406 (60.5)	84 (54.2)	78 (54.2) *	156 (67.8) *
TET	740 (89.7)	601 (89.6)	140 (90.3)	123 (85.4)	216 (93.9)
SXT	617 (74.8)	506 (75.4)	112 (72.3)	88 (61.1) *	195 (84.8) *
Total isolates	826	671	155	144	230

* Number and percentage of isolates resistant to antimicrobials with *p* < 0.01.

**Table 3 pathogens-12-00112-t003:** Percentages of isolates simultaneously resistant to n° antibiotics (from 0 to 10) and to ≥ 6 antibiotics (multi-resistant) from 2017 to 2021, listed by all the isolates and enterotoxigenic *Escherichia coli* (ETEC) F18+. Nalidixic acid (NA), amoxicillin and clavulanic acid (AMC), ampicillin (AMP), cefazolin (CZ), enrofloxacin (ENR), florfenicol (FFC), gentamicin (GEN), kanamycin (KAN), tetracycline (TET), trimethoprim + sulfamethoxazole (SXT).

% of Resistances in the Overall Strains								
		Year of Isolation					Statistical Analysis	
n° antibiotics	Total	2017	2018	2019	2020	2021	r	*p*
0	0.5	0.9	0	0.5	0.8	0	-	-
1	1.5	2.8	2.4	0.9	1.2	0.8	−0.90	<0.05 *
2	2.9	2.8	3.2	2.8	3.9	0.8	−0.45	>0.05
3	3.8	2.8	4.8	2.8	3.5	5.8	0.56	>0.05
4	6.3	6.5	6.4	4.6	7.8	5.8	−0.00	>0.05
5	8.6	8.4	6.4	7.8	8.9	11.7	0.74	>0.05
6	10.2	13.1	8.8	12.4	8.2	9.2	−0.59	>0.05
7	13.8	12.1	12.8	17.1	13.2	11.7	−0.04	>0.05
8	16.8	15	20.8	19.8	14.4	14.2	−0.39	>0.05
9	17.4	14	20	15.7	20.2	15	0.12	>0.05
10	18.3	21.5	14.4	15.7	17.9	25	0.38	>0.05
>6	76.5	75.7	76.8	80.6	73.9	75	0.38	>0.05
N° tested isolates	826	107	125	217	257	120		
% of Resistances of F18+ Strains								
		Year of Isolation					Statistical Analysis	
n° antibiotics	Total	2017	2018	2019	2020	2021	r	*p*
0	1.3	5	0	1.3	1.3	0	-	-
1	0.9	0	2.9	0	1.3	0	−0.20	>0.05
2	1.7	0	0	1.3	3.9	0	0.36	>0.05
3	0.4	0	0	1.3	0	0	0	>0.05
4	3	5	5.9	1.3	2.6	4.2	−0.42	>0.05
5	6.5	10	0	10.5	3.9	8.3	0.02	>0.05
6	10	20	5.9	11.8	10.5	0	−0.75	>0.05
7	13.5	5	8.8	17.1	15.8	8.3	0.41	>0.05
8	20.4	25	41.2	17.1	13.2	20.8	−0.53	>0.05
9	20.9	20	23.5	18.4	25	12.5	−0.44	>0.05
10	21.3	10	11.8	19.7	22.4	45.8	0.91	<0.05 *
>6	86.1	80	91.2	84.2	86.8	87.5	0.41	>0.05
N° tested isolates	230	20	34	76	76	24		

* the trend was considered statistically significant for *p* < 0.05.

## Data Availability

The data presented in this study are available in this manuscript and in Appendix A.

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
