# Peer review of "Antimicrobial Resistance and Virulence Factors Assessment in Escherichia coli Isolated from Swine in Italy from 2017 to 2021"

_pathogens, 2023, doi:10.3390/pathogens12010112_

Round 1

Reviewer 1 Report

Antimicrobial resistance and virulence factors assessment in 2 Escherichia coli isolated from swine in Italy from 2017 to 2021.

Thank you very much for this manuscript reporting Escherichia coli antimicrobial resistance and virulence in material originating from pigs.

The results are interesting and highly relevant for pig production and the one health perspective.

I have one overall comment, followed by some more specific comments. Especially the tables needs some editing for ease of understanding.

In general, I lack a ‘Materials and methods’ section (might be brief) that gathers and explains all information about the samples that are analysed, where do they come from, how were they kept, which methods were used, which laboratory did the analyses etc. Which hypotheses were tested? Statistical analyses, methods and programs used for the statistical calculations.

L. 18-20: “E.coli lacking virulence factor genes were more resistant to” … please add a compared to….

L. 24: The word ‘tested’ appears twice in this sentence.

L. 29: Please write out the abbreviation PNCAR, the first time it is mentioned

L. 36: is CM an abbreviation of coliform mastitis? Please write the word behind the abbreviation.

L. 34-38: This is a long sentence, please divide into two,  for better understanding.

L. 40: ETEC is an abbreviation of enterotoxigenic E.coli, please write this.

L 51-53: Please present the aims of this report in the same order as you present your results: Prevalence of resistance in E.coli towards antimicrobials, development over time and effect of virulence factors on AMR prevalence. Keep this order in all sections of the paper (Materials and Methods, Results, Discussion and Conclusion)  

L. 55: Do you have information whether pigs were treated with AM? Is it possible to report virotypes and resistance per diagnose (ND, PWD, septicemia, ED UTI etc)?

L 66 Please report both numbers and percentages, and do this consistently throughout the result and other sections: eg: 4/826 (0.5%)

L 96-100: This section is a bit difficult to follow. Please simplify and rephrase

L 119: should I be able to see this result in Table 4? Perhaps Table 4 needs a ‘Total’ column with a sum of results for all 5 years?

Table 1. α-hemolisin or –lysin? Layout: Please remove the horisontal lines in the table content below the headlines. Headlines in bold (also ‘Toxins’) . The table should be self-explanatory, so please explain the abbreviation either in table text or as footnotes. Consult: https://www.mdpi.com/authors/layout#_bookmark42

Table 2. Please consider the layout: I think you should place ‘n’ and ‘%’ above each column for ease of understanding.  Some text is in bold with a line, which must be an error?

The table text: Number and percentage of E. coli resistant to the tested antimicrobials. E. coli isolates tested in total (n = 826), virulence genes positive and negative isolates (n. 671, n. 155) and ETEC F4+ (n. 144) and ETEC F18+ (n. 230).

Table 3 + 4. What kind of correlation does the p values represent: An overall effect of time or a difference between 2017 and 2021 numbers? Layout: skip vertical lines. This table contains a lot of information: Is reporting of all F4 + F18 results essential or could these data be placed in supplementary material?  

L 111 Are you referring to the total percentage of resistant strains … or should the referral be to Table 3?

L 132-138: Since you are not reporting any results regarding the age of the pigs, I don’t find this part relevant for this paper and I suggest you delete it.

L142 : start your discussion with this part.

Reviewer 2 Report

Report

The present article pathogens-2098110- title: Antimicrobial resistance and virulence factors assessment in Escherichia coli isolated from swine in Italy from 2017 to 2021 aimed at investigating the antimicrobial susceptibility and some virulence genes of pathogenic E, coli recovered from pigs in Italy during the period 2017-2021. This study is important from a medical microbiology point of view. However, I have certain major comments that should be considered before accepting this work for publication, these include:

1.     The abstract should be rephrased to start with the background, aim, result, and final conclusion.

2.     Abbreviations should be written in full sentences in the first mention and the abbreviation should be used in the whole manuscript thereafter (Example: lines 17 and 19, ETEC F18+,  and ETECF4+. All abbreviations should be mentioned consistently in the whole manuscript). In line  29 (PNCAR) stands for what?? It should be written in a full sentence.

3.     In Line 23 “The P-value should be added after the word” Statistically significant”. And in line 24 (tested is repeated two times, the second one should be omitted and the sentence should be ended with “all the tested antibiotics”.

4.     The authors should provide the source/reference of the data displayed in Table 1.

5.     Result section, “2.1 Detection of the virulence genes” the author should figure out the results in a tabular form (as a table) and provide the number and% of each E. coli category for better understanding and for more clarification of the results.

6.     The author should give the rationale for selecting only the tested antibiotics, particularly Nalidixic acid is no longer used (more than 20 years ago, it was banded )in practice because of its toxicity, and instead new generations of quinolones are now in practice such as ciprofloxacin and levofloxacin??. The selection of antibiotics should be based on international guidelines such as CLSI or EUCAST guidelines.

7.     Table 4 should be moved to the results.

8.     The results of the tested toxins (LT1, STaP, STb, and Stx2e) are not displayed in the results.

9.     The authors measure the antimicrobial susceptibility only via quantitative means (Dis diffusion) which is not enough to judge the susceptibility of the collected isolates. Therefore, MIC measurement is important to confirm the results of the antimicrobials susceptibility???

10.  In material and methods, the authors did not provide any information about specimen collection, procession, recovery, and identification of the E. coli isolates???

11.  Conclusion section should be rewritten and summarized and be focused only on the obtained. results. (The author did not prove the transmission of traits from pigs to humans nor test the possibility of horizontal gene transfer and the presence of mobile genetic elements). So, this information should be removed from the conclusion.

12.  The author should test the clonal relationship of the obtained E. coli isolates either genetically (ERIC-PCR or RAPD-PCR) or phenotypically via the construction of the heatmap analysis of the tested isolates.

Reviewer 3 Report

The manuscript entitled "Antimicrobial resistance and virulence factors assessment in Escherichia coli isolated from swine in Italy from 2017 to 2021" described the antimicrobial resistance situation in E. coli isolated form pathological samples of swine in Emilia Romagna (Italy) from 2017 to 2021. The Authors has also revealed few virulence factors of E. coli and their relation with AMR. The manuscript has a positive impact on antimicrobial resistance research. However, the manuscript requires few corrections.

Line 13-14: Best to write- bacteria showed the highest level of resistance to Ampicillin (95.9%), tetracycline (89.7%), cefazolin (79.3%), and trimethoprim + sulfamethoxazole (74.8%).

Background and justification of the study is not well established. There is opportunity to improve the introduction section.

Materials and method section the study is too short to understand the AMR determination procedure. Sample type, number, sample collection and processing are absent. Swine pathological samples were taken for investigation. Authors should add sample wise the pathological conditions. I think, the isolation and AMR phenotype detection procedure need to be added in details. Please list the gene specific primers with references. In addition, add the PCR conditions.

Graphical presentation of susceptibility report is recommended for better understanding. Please add table 4 to results section.

Reviewer 4 Report

The author surveys 826 E. coli isolates for toxin and fimbrae genes. These isolates are sourced from swine in Italy from 2017 to 2021 using classic disc diffusion assays, multiplex PCR. 

The authors find that Ampicillin, tet, cefazolin, and sulfanomides were highly represented resistances and isolates harboring virulence genes resistant to various protein synthesis inhibitors. Relatively few were MDRs. (18%)

While these kinda of surveys are relatively straight forward, the work is critical to knowing trends in our food supply. 

The finding that use of some antibiotics is associated with increased prevalance of resistance is in-line with findings from others. I particularly appreciate the discussion about relevant antibiotics and their intersection with their use in husbandry but would like to see this discussion deepened. 

This paper is also well-wrtten and had appropriate methodology

Round 2

Reviewer 1 Report

Thank you very much for this second draft. The understanding of the paper and its results have improved considerably - well done!

I only have a few minor comments: 

Table 3: Missing part/text? 

L 329 E.coli should be in italics

L 363: both p<0.05 and p< 0.01 are reported in Tables 2 and 3

Author Response

#Reviewer 1

Comments and Suggestions for Authors

Thank you very much for this second draft. The understanding of the paper and its results have improved considerably - well done!

R: The authors agree with the considerable enhancement and we are sincerely grateful to the Reviewer for the reviews and suggestions that contributed to improve the manuscript.

I only have a few minor comments: 

Table 3: Missing part/text?

R: the missing text was reported in Table S2, the remined empty lines have been deleted

L 329 E.coli should be in italics

R: Correction done

L 363: both p<0.05 and p< 0.01 are reported in Tables 2 and 3

R: The authors corrected the Tables according to the p value reported in the material and method section (4.5 statistical analysis).

Reviewer 2 Report

The authors addressed the required comments and suggestions and now the manuscript is suitable to be published

Author Response

#Reviewer 2

The authors addressed the required comments and suggestions and now the manuscript is suitable to be published

R: The authors thank the Reviewer for the reviews that contributed to improve the manuscript.

Reviewer 3 Report

The quality of the manuscript has been promoted now. Authors should check spelling, space, and others for error free manuscript.

Author Response

The quality of the manuscript has been promoted now. Authors should check spelling, space, and others for error free manuscript.

R: Authors are sincerely grateful for the positive comment of the Reviewer, a check has been done before uploading the revised manuscript.